# Experimental investigation into the effect of compliance of a mock aorta on cardiac performance

Katie Cameron[1], Mouhammad El Hassan[2,3], Reza Sabbagh[3], Darren H. Freed[4], David S. Nobes[3]*

1 Department of Biomedical Engineering, University of Alberta, Edmonton, Canada, 2 Prince Mohammad Bin Fahd University, Al-Khobar, Saudi Arabia, 3 Department of Mechanical Engineering, University of Alberta, Edmonton, Canada, 4 Departments of Surgery, Physiology & Biomedical Engineering, University of Alberta, Edmonton, Canada

* dnobes@ualberta.ca

**Data Availability Statement:** All relevant data are within the manuscript.

**Funding:** Financial support for this project was provided by the Natural Sciences and Engineering

## Abstract

Demand for heart transplants far exceeds supply of donated organs. This is attributed to the high percentage of donor hearts that are discarded and to the narrow six-hour time window currently available for transplantation. Ex-vivo heart perfusion (EVHP) provides the opportunity for resuscitation of damaged organs and extended transplantation time window by enabling functional assessment of the hearts in a near-physiologic state. Present work investigates the fluid mechanics of the ex-vivo flow loop and corresponding impact on cardiac performance. A mechanical flow loop is developed that is analogous to the region of the EVHP system that mimics in-vivo systemic circulation, including the body's largest and most compliant artery, the aorta. This investigation is focused on determining the effect of mock aortic tubing compliance on pump performance. A custom-made silicone mock aorta was developed to simulate a range of in-vivo conditions and a physiological flow was generated using a commercial ventricular assist device (VAD). Monitored parameters, including pressure, tube distension and downstream velocity, acquired using time-resolved particle imaging velocimetry (PIV), were applied to an unsteady Bernoulli analysis of the flow in a novel way to evaluate pump performance as a proxy for cardiac workload. When compared to the rigid case, the compliant mock aorta case demonstrated healthier physiologic pressure waveforms, steadier downstream flow and reduced energetic demands on the pump. These results provide experimental verification of Windkessel theory and support the need for a compliant mock aorta in the EVHP system.

## Introduction

Heart transplantation is the gold standard treatment for end-stage heart failure patients but demand for viable donor hearts far outweighs supply [1,2]. This critical shortage is most often attributed to three factors: high discard rate amongst available donor hearts [1,3], the limited time window of six hours available for transplantation using the current standard of care [4]

Research Council (NSERC) of Canada (DSN),
Canadian Foundation for Innovation (DSN),
Canadian National Transplant Research Program
(DHF), and University Hospital Foundation (DHF).

**Competing interests:** The authors have declared
that no competing interests exist.

(hypothermic static storage), and the non-utilization of hearts from extended criteria donors (ECD) and donation after circulatory death (DCD) donors [5,6]. To reconcile the demand for heart transplants with supply, the donor pool must be expanded.

In an effort to expand the donor pool, *ex-vivo* heart perfusion (EVHP) has been proposed as a method for resuscitating and preserving damaged donor hearts from the time between donation and transplantation [7–9]. Previous EVHP technologies have perfused the heart in Langendorff mode, where retrograde flow is supplied to the aortic root [2]. While clinical trials have shown that this method provides better recipient outcomes than does hypothermic static storage [10,11], recent literature cautions that the ability of this technology to contribute to donor pool expansion will be limited by its reliance solely on metabolic markers to assess transplantation viability [3,5].

Advancements are currently being made in EVHP technology that enables a near-physiologic working mode [1,12–19], wherein flow is supplied to the ventricles so they can fill and eject perfusate, a mixture of blood and support nutrients [17], at normothermic temperature (37 ˚C) as would occur in the body. Since the ventricles are doing mechanical work in the same fashion as they would *in-vivo*, both metabolic and mechanical conditions can be controlled in response to monitored changes in cardiac performance [1,12–19]. This provides transplant teams with more functional information that should empower them to make better decisions about transplantation viability [2–5]. Recent literature published on this EVHP system concluded that the assessment of functional parameters, such as left ventricular stroke work, was indeed the most reliable means for evaluating organ health [1]. Control of the metabolic environment required to keep the heart healthy in the proposed EVHP system is well-understood and continues to be optimized [1,12–19] but the fluid mechanics of the system and its corresponding impact on cardiac performance have yet to be studied prior to this investigation.

The most complex flow region in the system is the left flow loop which simulates the *in-vivo* systemic circulatory system. This *in-vivo* region is most notably characterized by a complex fluid-structure interaction of the highly compliant aorta responding to the significant unsteady effects of the pulsatile cardiac cycle [20]. Aortic compliant response is not currently present in the proposed EVHP system because the aortic root of the explanted donor heart is connected directly to rigid tubing during installation of the heart into the system [1]. Based on the widely held understanding that aortic compliance is an important marker for cardiac performance and overall cardiovascular health *in-vivo* [21], it is possible that the lack of elastic response in the proposed EVHP system is negatively impacting cardiac performance when the system is being run in working mode. However, this impact has not been quantified and, to the authors' knowledge, adding a compliant mock aorta to an EVHP system capable of near-physiologic working mode has not been investigated prior to this experimental study. While there is research in the literature that uses metrics such as hydrodynamic efficiency to assess the physiological conditions of the heart [22], there is limited research into how the circulatory conditions affect heart performance in an ex-vivo setup.

For this reason, this work has undertaken the development of a mechanical flow loop analogous to the left side of the proposed EVHP system to study the effects of a compliant mock aorta on the aortic and arterial pressure waveforms as well as on the arterial velocity fields, which were obtained using time-resolved particle imaging velocimetry (PIV). The aortic pressure data and high resolution arterial velocity data were then used to compare the energy demands on the pump under rigid conditions to those under compliant conditions. This determination of energy output of the pump during systolic ejection was used as a metric for assessing the impact of a compliant mock aorta on cardiac performance under a variety of experimental conditions, which was the goal of this study. The work is being done in the

interest of advancing both EHVP technology and experimental understanding of the fluid mechanics involved in aortic compliant response.

## Background

The well-defined mechanical contraction-relaxation process that the heart undergoes to circulate blood is known as the cardiac cycle [23]. Information about the cardiac cycle, as well as assessment of cardiac performance, is most commonly understood through interpretation of the pressure waveforms in the peripheral arteries, those that are far from the heart [24]. While this is common clinical practice due to the non-invasive nature of such a measurement, more important information can be obtained from central aortic pressure waveforms that are obtained directly from the aorta [25]. Understanding the values that can be obtained from central aortic pressure waveforms provides a foundation upon which to study cardiovascular flow.

A labelled diagram of a generic central aortic pressure waveform [23] is shown in Fig 1. The cardiac cycle is defined by two phases: systole, which characterizes ventricular contraction and diastole, which characterizes ventricular relaxation. During systole, which ideally comprises approximately one third of the cardiac cycle [23], the left ventricle is contracting and ejecting blood into the aorta. Systolic pressure ($P_s$) is defined as the peak pressure achieved during this ejection phase. Once this peak pressure is reached toward the end of the contraction of the left ventricle, there is a gradual decrease in pressure in the aorta. Upon closing of the aortic valve, there is a momentary rise in aortic pressure due to the elastic recoil of the aortic wall against the closed valve [23]. This is referred to as the dicrotic notch on the pressure waveform. Once

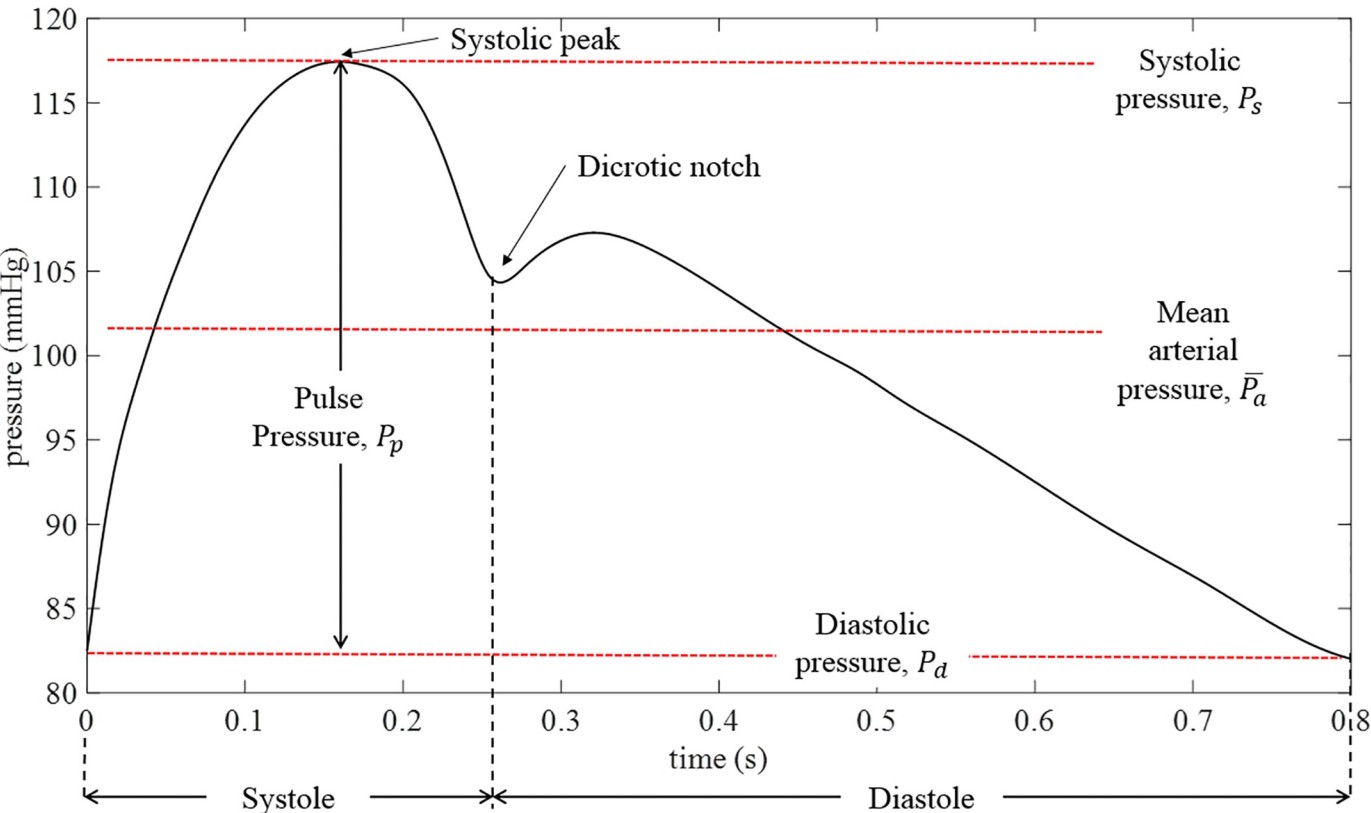

**Fig 1. Labelled diagram of a generic central aortic waveform.**

the valve is closed, the left ventricle begins to relax, marking the beginning of diastole. During diastole, aortic pressure decreases as blood is delivered throughout the arterial system. The minimum pressure in the cycle, is defined as diastolic pressure ($P_d$). The difference between $P_s$ and $P_d$ is defined as the pulse pressure ($P_p$) which represents the force generated by the left ventricle during ejection. Mean arterial pressure ($-P_a$) is the mean pressure over one cardiac cycle, often calculated by integrating the central aortic pressure waveform over the cycle time.

The theory behind pulsatile flow is well-understood [20,26–28]. A method for quantifying the velocity and flow rate of a pulsatile flow in a straight, rigid circular pipe subjected to a known unsteady pressure gradient was first published by J.R. Womersley [26] in 1955. At that time, the non-dimensional Womersley number, $\alpha$, was defined to describe the relationship between unsteady and viscous effects in a pulsatile flow regime [26] as:

$$\alpha \;=\; \frac{D}{2}\sqrt{\frac{\rho\omega}{\mu}} \tag{1}$$

where $D$ is the tube diameter (m), $\rho$ is the fluid density (kg/m$^3$), $\omega$ is the angular frequency (rad/s) of the pulse wave given by the frequency of pulsation ($f$) as $\omega = 2\pi f$ and $\mu$ is the fluid dynamic viscosity (Pa.s). It has been suggested [29] that for $\alpha = 1$ a limit is reached where fluid behavior begins to vary significantly from the quasi-static case. In human aortas, $\alpha$ ranges from approximately 12–20 [20,30].

In high $\alpha$ flows where the fluid is assumed to be incompressible, conservation of energy in the system can be described as:

$$p_1 + \rho g z_1 + \frac{\rho u_1{}^2}{2} + \rho\int_0^1 \frac{\mathrm{d}u}{\mathrm{d}t}\,\mathrm{d}s \;=\; p_2 + \rho g z_2 + \frac{\rho u_2{}^2}{2} + \rho\int_0^2 \frac{\mathrm{d}u}{\mathrm{d}t}\,\mathrm{d}s + p_{losses} \tag{2}$$

where $p_i$ can be the pressure at two points in the flow $\{i = 1 \text{ or } 2\}$ (Pa) along a streamline; $g$ is the gravitational constant 9.81 m/s$^2$; $z_i$ is the elevation of point $i$ (m); $u_i$ is the average velocity at point $i$ (m/s); and $\frac{\mathrm{d}u}{\mathrm{d}t}$ is the flow acceleration being integrated along a streamline, ds; $p_{losses}$ are the frictional losses that occur along the tube length (Pa), and $L$ is the length of the fluid volume being accelerated (m). Previous investigations [27,31,32] have applied this equation to cardiovascular analog flow regimes by relating the pressure gradient driving the flow to the resulting temporal and convective accelerations. The approximation of neglecting the unsteadiness and the friction losses have been criticized in [33]. A similar approach is undertaken here to calculate the pressure drop. Using the assumptions that the height change between the points has a negligible effect on the flow and that the tube has a uniform diameter between points 1 and 2 and using the assumption that the acceleration varies with area in the same way as $u$ [32], the pressure differential driving the flow is described as:

$$\Delta p(t) - p_{losses} \;=\; \rho\int_1^2 \frac{\mathrm{d}u}{\mathrm{d}t}\,\mathrm{d}s + \frac{1}{2}\rho(u_2{}^2 - u_1{}^2) \tag{3}$$

$$\Delta p(t) - p_{losses} \;=\; \rho L\frac{\mathrm{d}u_2}{\mathrm{d}t} + \frac{1}{2}\rho(u_2{}^2 - u_1{}^2) \tag{4}$$

where $\Delta p(t)$ is the time-varying pressure between the two locations in the system (Pa).

In addition to analytical solutions for the pulsatile flows obtained after applying simplifying assumptions, there have been many numerical and experimental investigations into pulsatile flows through rigid tubes [34–37], but these are ultimately limited in their physiological relevance because *in-vivo*, blood interacts with compliant arteries. This compliant response is particularly pronounced in the aorta which converts some of the ejection

energy into elastic energy through expanding its size by 20–60% during systole [38] and storing up to 50% of the ejected stroke volume [39]. The amount of expansion is determined by the intrinsic and operative distensibility of the tissue [40] as well as the difference between the pulse pressure generated by the left ventricle and the pressure surrounding the aorta, the transmural pressure. Although the physical meaning of distensibility is the expansion percent per mmHg of internal fluid pressure, the literature most commonly represents it in units of mmHg$^{-1}$ and as such, that unit is adopted in this paper. Typical aortic distensibility values range from approximately $(1.33–8.90) \times 10^{-3}$ mmHg$^{-1}$ [41,42] depending on a person's age and overall health.

The compliant response of the aorta has a significant impact on wave propagation effects *in-vivo* and therefore is widely recognized as being a clinically important marker for overall cardiovascular health [43–46]. There have been several clinical investigations into wave propagation effects *in-vivo* [47], and experimental investigations into flow fields through compliant phantoms in low $\alpha$ regimes [48–50], but, to the authors' knowledge, there has been little experimental investigation into controlled changes in aortic compliance within a physiological range in an *ex-vivo* environment. There has been a great deal of research into understanding velocity distributions through compliant phantoms using PIV or particle tracking velocimetry (PTV) in cardiovascular flow regimes [38,48,49,51], including the effect of certain pathophysiology such as stenosis [52,53], in compliant pulsatile flow regimes through the use of stereo-PIV and time-resolved PIV. However, to the authors' knowledge, there remains limited understanding of the effect of physiologically relevant aortic compliant response on downstream velocity fields and pump performance. The approach presented in this study of evaluating the energy requirements of a cardiovascular analog pump using central aortic pressure waveforms and high resolution downstream velocity field data is novel.

## The current EVHP system

To properly pose the investigation, it is important to understand the current embodiment of the proposed EVHP system upon which the analog system used for this experimental investigation is based. A schematic of the EVHP system, including both the heart and flow loop is shown in Fig 2. To maintain the heart in near-physiologic working mode, the EVHP system circulates perfusate at normothermic temperature through two parallel flow loops that mimic the *in-vivo* systemic and pulmonary circulatory systems. The system is comprised of a pacemaker-stimulated heart, a reservoir, an arterial filter, two centrifugal pumps, an oxygenator and a tubing network.

In Fig 2, the heart is connected to the tubing network via four connection points: the left atrium (LA), the aortic root just above the left ventricle (LV), the right atrium (RA) and the pulmonary artery just above the right ventricle (RV). Pump 1 (P1) supplies perfusate to the LA and RA, the heart's receiving chambers, which contract upon pacemaker stimulation. Upon artial contraction, perfusate flows into the LV and RV which subsequently contract according to the sinus rhythm set by the pacemaker. Perfusate is pumped out of the LV and RV into ⅜" and ½" tubing in the mock systemic and pulmonary circulatory loops, respectively. Pump 2 (P2) provides an afterload to the heart, simulating the resistance of the arterial system by supplying a constant flow against the direction of ventricular ejection. This provides the back pressure required to close the aortic valve (AV) and supply coronary perfusion during diastole. Flow ejected from the left ventricle combines with flow from P2, passes through an oxygenator, then combines with flow from the right ventricle and finally, returns to the reservoir. Pressure and flow monitoring at several locations, shown in Fig 2, provide direct feedback of hemodynamic conditions and cardiac performance.

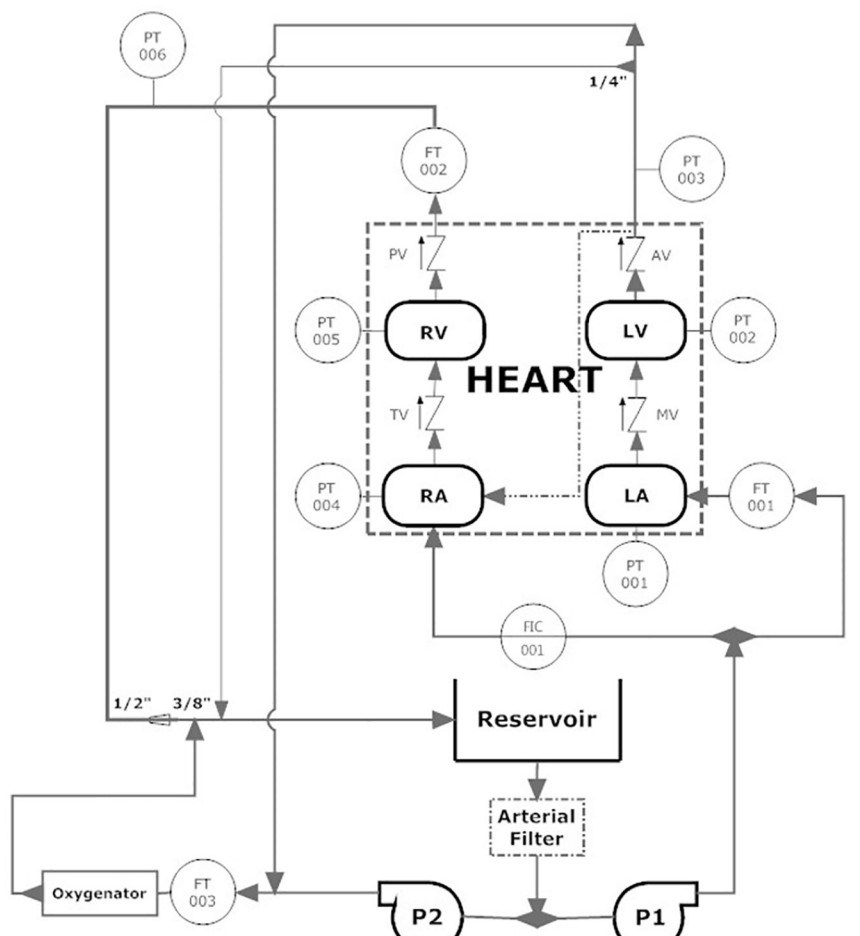

**Fig 2. Operating schematic of the current EVHP system.**

## Methodology

The aim of this investigation is to determine the impact of mock aorta tubing compliance on pump performance in an EVHP analog flow loop. To this end, a mechanical flow loop with a custom-made mock aorta and a downstream imaging section were constructed to mimic the left flow loop of the EVHP system. Pressure waveforms were monitored and velocity fields obtained using time-resolved PIV under a variety of operating conditions using water as the working fluid. The pressure data and high-resolution velocity data were used to evaluate pump performance in terms of energy output by the pump during systolic ejection. Results obtained using the mock aorta were compared to those obtained from the rigid tubing cases subjected to the same operating parameters.

### Mock aorta development

To achieve the desired physiological range of compliant response, a mock aorta, shown in Fig 3, was designed and molded in-house. The tube included a flange to facilitate connection to the rigid part of the flow loop. Previous investigations have established that silicone rubbers can demonstrate elastic responses similar to those of human arteries under physiological pressure conditions [54]. Off-the-shelf silicones are commonly used to mimic small arterial

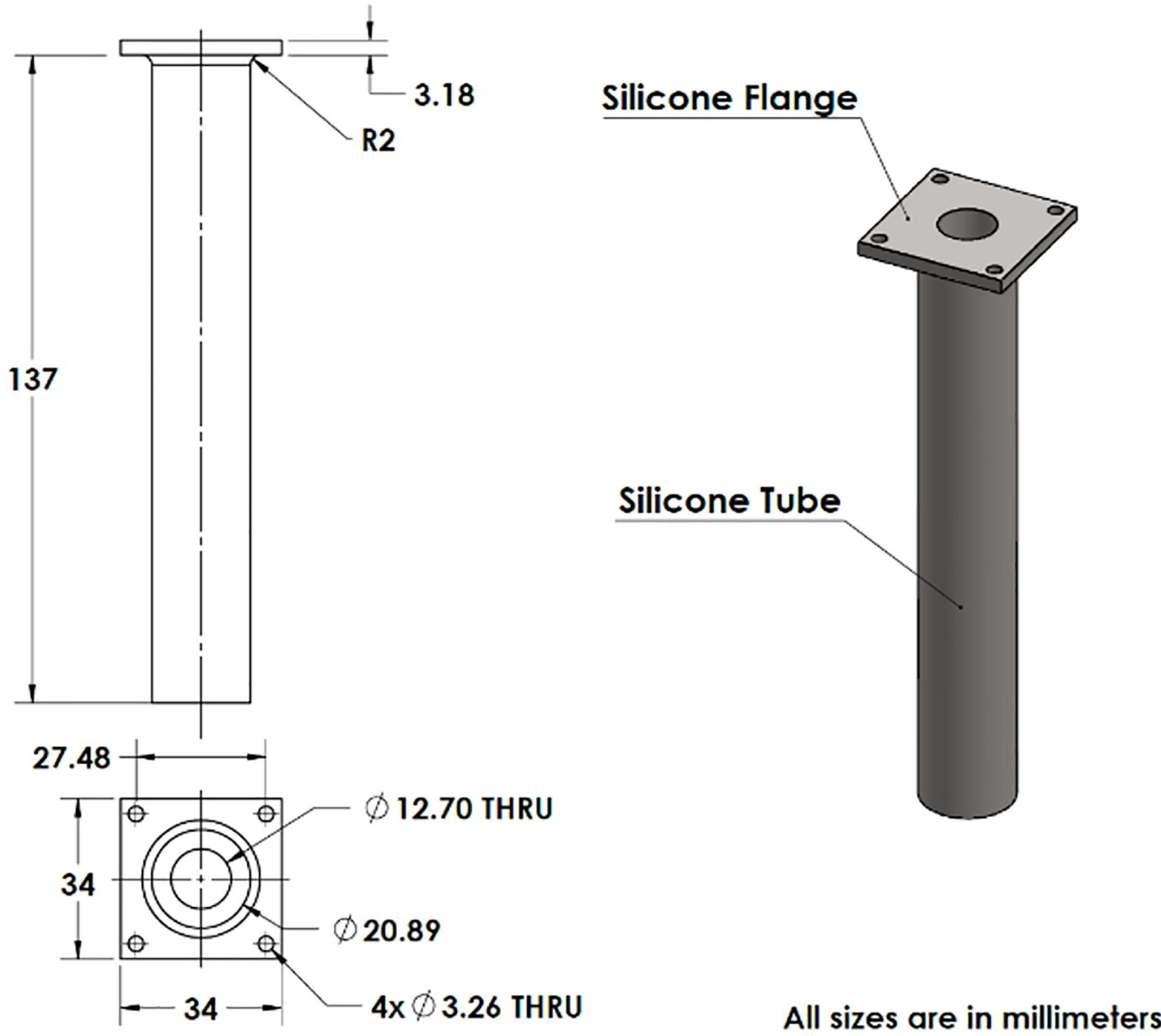

**Fig 3. Mock aorta 2D drawings and 3D isometric view.**

properties but more compliance was required for this investigation to capture *in-vivo* aortic response, so a two-part silicone (Ecoflex® 00–50, Smooth-On Inc.) was chosen. The silicone has a specific gravity of 1.07 g/cc and an elastic modulus of 12 psi [55,56].

The wall thickness of the tube is 4.1 mm. This thickness was chosen to accommodate a maximum of 80% distension under a maximum expected pressure differential from the pump used to simulate the heart.

## Experimental setup

The flow loop used for this experiment, shown in Fig 4 has two regions of interest. The first is the test section which was comprised of the mock aorta and pressure chamber for the

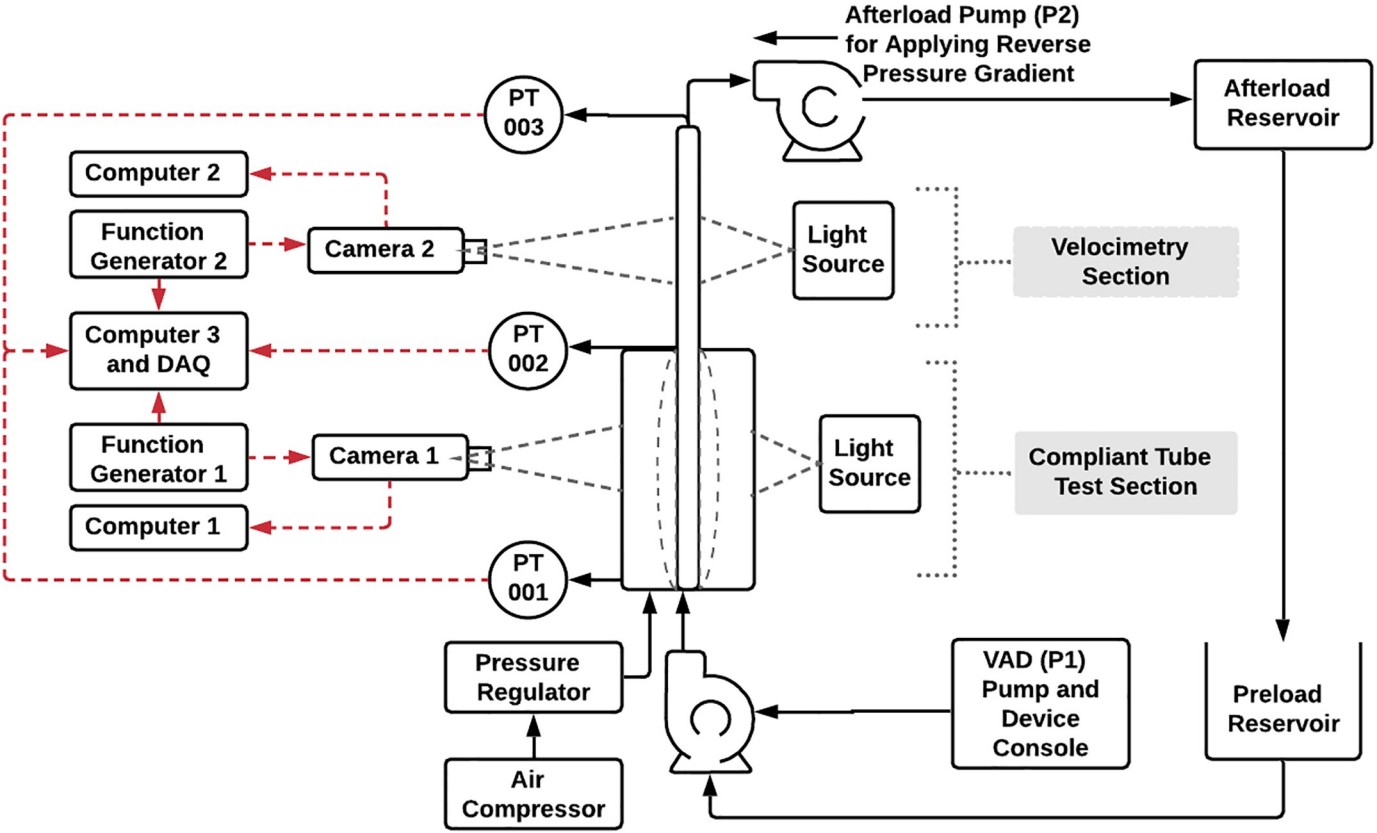

**Fig 4. A schematic of the experimental setup used to mimic the left flow loop of the EVHP system.**

compliant case and a ½" ID (inner diameter) plastic tube for the rigid case. The second region, the imaging section, was a ½" ID thin-walled glass tube surrounded by an imaging chamber comprised of four 1/16" thick acrylic windows. This region was the location where the down-stream velocity fields were obtained. The desired physiological flow was generated using a commercial ventricular assist device (VAD) (Thoratec® Corporation) denoted as P1 in Fig 4. A VAD is a pumping device composed of a sac and two prosthetic heart valves, one inlet and one outlet valve, that is commonly used in heart failure patients to bypass left ventricular pumping function [57]. The filling and ejection of the sac was controlled by a console with a fixed driving pressure in this case. When the driving pressure was applied, the sac compressed causing the inlet valve to close and the outlet valve to open, resulting in flow being ejected into the mock aorta. Subsequently, a vacuum pressure was applied to re-expand the sac, open the inlet valve and, in conjunction with the back pressure generated by the afterload pump (P2), close the outlet valve. As a result of the VAD driving pressure being fixed in these experiments, the pulse frequency and therefore cycle time varied depending on the afterload conditions of the system which were primarily regulated by back pressure generated by the afterload pump. The experimental setup mimics the left flow loop shown in Fig 2 (the flow path between the left ventricle and the reservoir). The notable differences between the experimental setup and the EVHP system shown in Fig 2 are the utilization of separate preload and afterload reser-voirs, using the VAD instead of a heart and the absence of a preload pump for the VAD. The preload pump has not been used in the experimental setup since there is no need to sustain

constant left atrial pressure such as what is needed in the EVHP system. Changing the pulse frequency from 1 to 2.33 Hz results in Womersley numbers that vary from 16 to 24.

An imaging system was developed to monitor the mock aorta tube compliance response throughout the pulse cycle. Camera 1 (Basler Pioneer[®], Basler Vision Technologies) with a resolution of 648×488 pixels was used to acquire images of the mock aorta over multiple pump cycles at a collection rate of 210 fps. This was taken in the central region of the test section that included a pressure chamber ($P_{ch}$) to maintain a controlled background pressure and viewing window. A pressure regulator with a manual dial was used to control the chamber pressure.

Downstream velocity fields were obtained using Camera 2 (Phantom[®] v611, Vision Research, Inc.), with a resolution of 1280×800 pixels that captured 1054 images at a collection rate of 1000 fps. The flow was seeded with 30–50 μm solid glass spheres (Spheriglass 3000, Potters Industries, Inc.). The average particle size was approximately 2.5 pixels captured over a focal plane thickness of approximately 0.45 mm. Images were collected in back-illumination/ shadowgraph mode using a high current green 4"×4" side-fired LED backlight. Pressure monitoring (Edwards[®] Truwave Disposable Pressure Transducers) was available at the inlet (PT001) and outlet (PT002) of the mock aorta as well as at the outlet of the imaging section (PT003) to allow for analysis of central aortic and downstream pressure waveforms. Pressure data was collected at a sampling rate of 4 kHz to ensure the key waveform features were clearly captured.

The two experimental parameters that were adjusted to control the experiment were the afterload pump speed ($\omega_{Al\ pump}$) and, for the compliant mock aorta case, the chamber pressure ($P_{ch}$). Experiments were performed at two values of $P_{ch}$, 103 and 155 mmHg, and 4–5 settings of $\omega_{Al\ pump}$ depending on how far the tube could distend before contacting the chamber walls. Varying $\omega_{Al\ pump}$ altered the afterload conditions of the system, which impacted systolic ejection time and pulse pressure, while varying $P_{ch}$ facilitated control over the magnitude of compliant response. The afterload pump speeds were chosen to produce tube expansion within physiological range of aorta expansion during systole (120–160%). It therefore resulted in achieving aorta expansion of 130–156% in the experiments.

Pressure waveforms, mock aorta images and the camera trigger signals were collected using an in-house developed code (LabWindows CVI, National Instruments) and coupled such that tube positions and downstream velocity results could be correlated with specific locations on the pressure waveforms. Pressure data obtained from PT001, was filtered using zero-phase digital filtering with a cut-off frequency of the 10[th] harmonic of the raw data. A frequency range of 10 harmonics was chosen in accordance with literature indicating that 6–15 harmonics are adequate to distinguish the features of an aortic waveform [58–62].

Images obtained from Camera 1 were processed with an in-house developed intensity peak detection code. The code tracked the locations of the tube walls by finding the locations of peak intensity changes in each image and using the distance between these two peaks to calculate the tube diameter in each image. Performing this processing technique on each captured image allowed monitoring of mock aorta diameter variations across multiple pump cycles.

Images obtained from Camera 2 were processed using commercial PIV software (DaVis 8.1.4, LaVision GmbH). First, the images were inverted (black/white) and a geometric mask was applied to constrain the image to the desired field of view. Decreasing multi-pass time series cross-correlation was used to interrogate the image to determine the vector map. The first three passes were 256×256 square windows with 50% overlap and the last three passes were 32×32 with 4:1 ellipsoid windows with 50% overlap. A large window size was necessary for the first three passes to capture the high ejection velocities from the VAD, and the ellipsoid shape was used in the final passes to improve spatial resolution in the near-wall region. After applying the processing scheme, vector maps were obtained during systole and diastole, an

example of which is shown in Fig 5. The vector maps are normalized by the width of the channel in both the radial and axial directions and are displayed from $x/D$ = 0 to 0.5 in the axial direction. The three plots in Fig 5 are for the initial stage of the systole when the flow begins to accelerate, Fig 5(a), at the develop phase of the systole around the maximum velocity, Fig 5(b) and for the diastole where the average centerline velocity is approaching zero, Fig 5(c). The example vector data set in Fig 5(c) illustrates the presence of secondary flows that arise at the end of diastole when the flow is nearing the end of deceleration. It should be noted that the systole phase Fig 5(a) corresponds to higher velocities on the lateral regions of the mechanical valve as compared to its central region. This can be related to the flow behavior of the 3 jets of the bi-leaflet valve which is also affected by the leaflets orientation as discussed in [63].

The average size of the particle image was 2.5×2.5 pixels$^2$, which is adequately resolved according to Prasad et al. [64] with the absence of the peak locking phenomenon [65]. The theoretical analysis of Westerweel [65] was used to estimate the error of the particle-image displacement. The accumulation of the error ratio and the bias error ratio gives the total error of ~1.4%. The maximal displacement errors are equal to 0.8% and 2.0% for the longitudinal and radial directions, respectively.

Based on the evaluation of the global bias error and the statistical uncertainty related to the data scattering around the mean values, the uncertainty of the PIV measurement is estimated to be in the range of ±0.2% to ±0.8% for the velocity components, and in the range of ±1% to ±4% for the corresponding rms velocity [66–68].

## Pumping performance

The vector arrays produced by the PIV code were imported into custom plotting code (Matlab, The Mathworks Inc.) to obtain centreline velocity values. Vector values were averaged over a region to prevent spatial variations from having a significant impact on the centreline velocity calculations. Averaging centreline velocities was undertaken over a region that spanned the full axial length of the image and a region either side of the centreline for $r/D$ = 0.413 to 0.588. There was often the presence of secondary flows as shown in Fig 5 and therefore non-uniformities in the axial direction occurred during diastole.

The values obtained from processing pressure and velocity data were used to calculate the total fluid energy during systole as a novel method for evaluating pump performance. A form of the energy equation for incompressible flow was used to relate the work done by the pump to the total energy in the fluid in the region of PIV data collection, defined as region 2, during systolic ejection such that:

$$E_{VAD} = E_{Static\ pressure} + E_{Dynamic\ pressure} + E_{Hydrostatic\ pressure} + E_{Temporal\ acceleration} \tag{5}$$

$$E_{VAD} = \frac{1}{\rho}\int_0^{t_s} \dot{m}_2(t)p_2(t)dt + \frac{1}{2}\int_0^{t_s}\dot{m}_2(t)[\bar{v}_2(t)]^2 dt + gz_2\int_0^{t_s}\dot{m}_2(t)dt + L\int_0^{t_s}\dot{m}_2(t)\left(\frac{dv}{dt}\right)_2 dt$$

where $E_{VAD}$ is the energy imparted by the pump to the fluid per systolic ejection (J/ejection), $p_2(t)$ is the instantaneous pressure obtained in region 2 (Pa), $-v_2(t)$ is the instantaneous velocity averaged across the tube section in region 2 obtained from PIV measurements (m/s), $z_2$ is the elevation of region 2 above the VAD outlet valve (m), $L$ is the height of the column of fluid being accelerated, which in this case is the height difference between the VAD outlet valve and point 2, $t_s$ is the systolic ejection time (s) and $\dot{m}_2(t)$ is the mass flow rate through region 2 (kg/s) at time $t$. This was calculated by integrating the mean velocity profile obtained from each PIV image in the imaging section where $D$, the constant size tube diameter (m) is used to determine volume flow rate and then multiplying by density to determine the mass flow rate

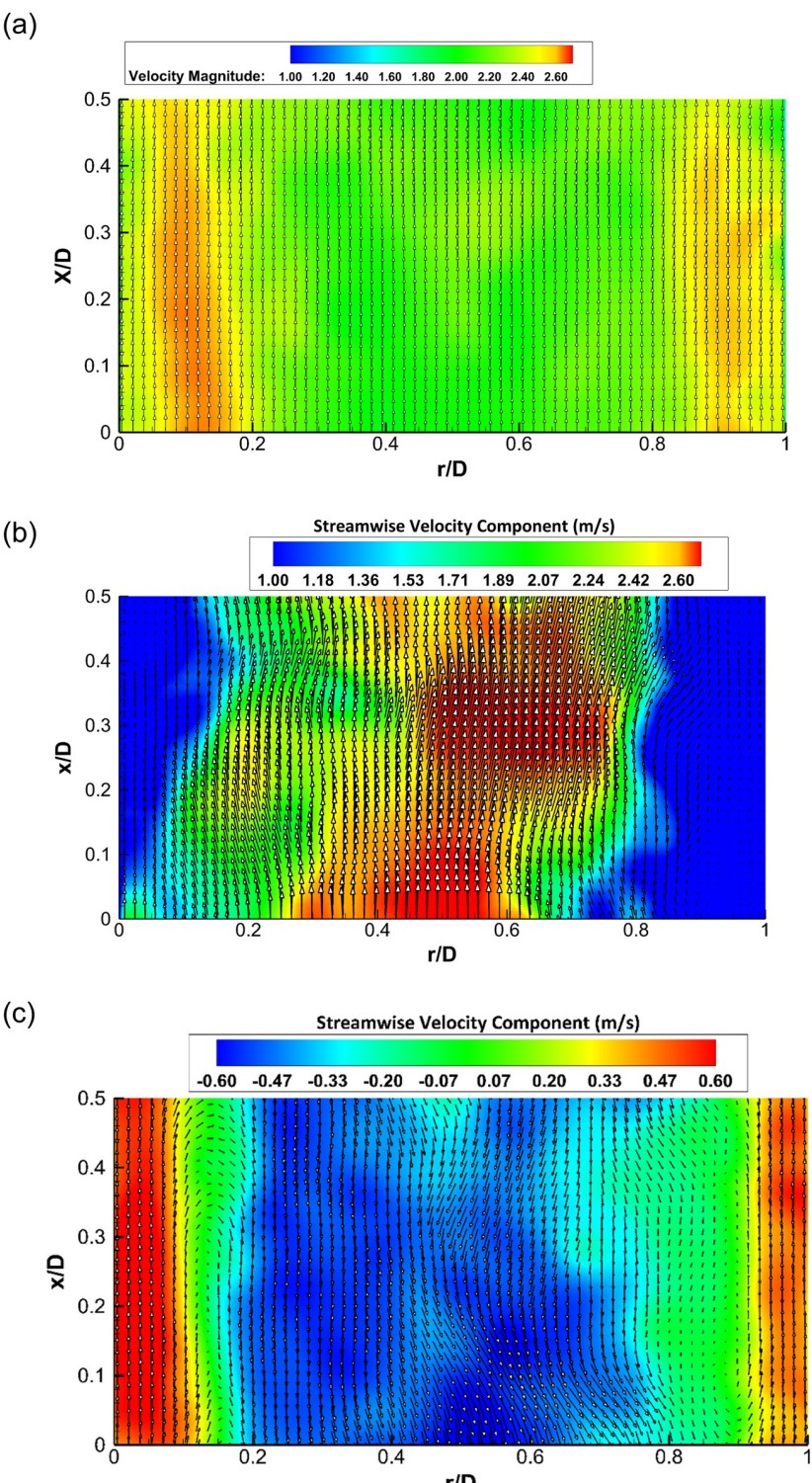

**Fig 5. Vector field images with a background color map of streamwise velocity component obtained from processing Camera 2 images, (a) at the beginning of systole, (b) during systole, (c) end of diastole phase.**

using:

$$\dot{m}_2(t) \;=\; \rho \int_0^{D/2} \int_0^{2\pi} v(r,\ t)\, r\, d\theta\, dr \;=\; 2\pi\rho \int_0^{D/2} v(r,\ t)\, r\, dr \qquad (6)$$

## Results & discussion

The results outlined in this paper include the presentation of pressure, tube distension and downstream velocity results obtained from one set of operating conditions, where $P_{ch}$ = 103 mmHg and $\omega_{Al\ pump}$ = 1235 RPM. In general, the description of features in the results were similar to other conditions investigated. As well, results regarding the impact of mock aorta distensibility on the transmission of flow pulsatility downstream and on pump work are presented.

### Pressure response

Fig 6 compares the pressure waveforms, $P$ obtained at the end of the rigid and the $P_{ch}$ = 103 mmHg mock aorta case with $\omega_{Al\ pump}$ = 1235 RPM over three pump cycles normalized by

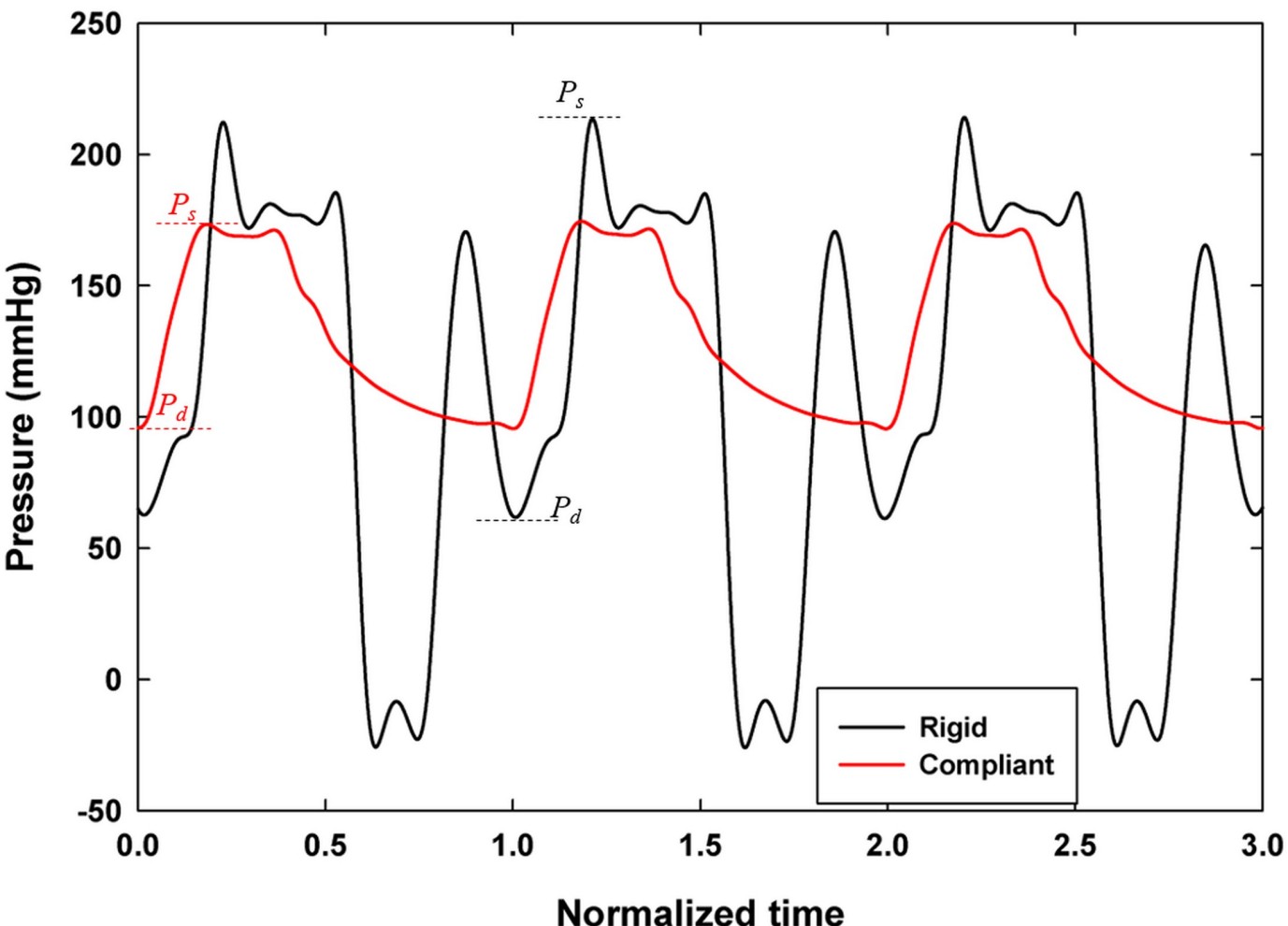

**Fig 6. Plot of central aortic pressure waveforms for rigid ($\omega_{Al\ pump}$ = 1235 RPM) and compliant mock aorta ($\omega_{Al\ pump}$ = 1235 RPM, $P_{ch}$ = 103 mmHg) cases over three normalized pump cycles. $P_s$ and $P_d$ locations are shown for rigid and compliant cases with the same colour codes.**

cycle time, τ. Data is presented as solid lines rather than discrete points due to the high sampling frequency (4 kHz) at which the data was collected. The waveform shape in Fig 6 is consistent across multiple cycles for both cases, but the two waveforms have different shapes and magnitudes. Fig 6 also shows that the incident and reflected waves are well-defined and that the dicrotic notch is visible for compliant tube at approximately $t/\tau = 0.4$. The rigid waveform has several peaks, indicating that there are multiple reflected waves involved in the system response. This is in contrast to the compliant case, which has two peaks—one incident wave and one reflected wave, as expected in a physiologic central aortic waveform. For the compliant case, the diastolic, systolic and mean arterial pressures are 91.7, 169 and 126 mmHg, respectively and for the rigid case, they are 68.1, 215 and 134 mmHg, respectively. The significantly higher systolic pressure suggests substantial augmentation of systolic pressure by the reflected waves during early systole. The large valley in the rigid pressure wave is most likely caused by forceful reflected waves impeding the VAD outlet valve's closure.

## Mock aorta compliant response

The distension response of the mock aorta is displayed in Fig 7 for $P_{ch} = 103$ mmHg case with $\omega_{Al\ pump} = 1235$ RPM over one and three normalized pump cycles. The two contributors to the tube expansion are shown in Fig 7; the tube distension resulting from the back pressure generated by the afterload pump alone (dashed line), and the distension resulting from $P_p$ (circles). The profile and magnitude of the response is consistent across multiple cycles. During diastole, the afterload pump distends the tube by 11.1%. During systolic ejection, the tube expands

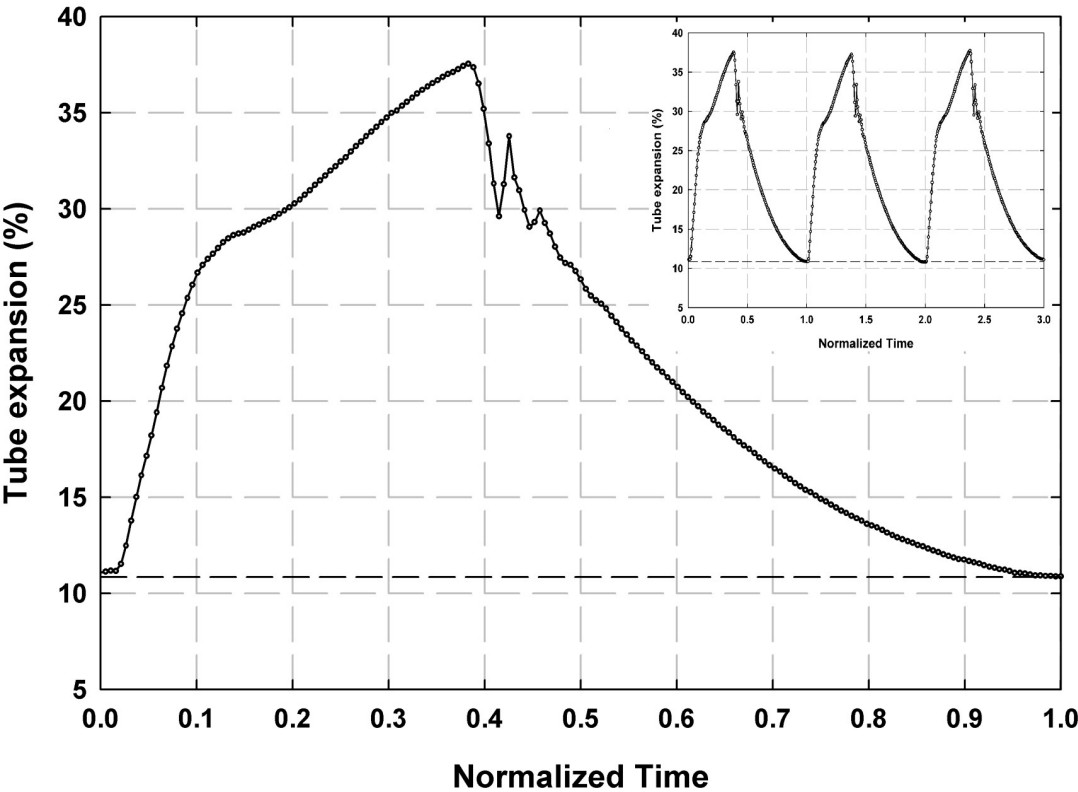

**Fig 7. Plot of mock aorta pulse pressure tube distension (circles) and afterload tube distension (dashed horizontal line) over one and three normalized pump cycles.**

quickly by approximately 28%, then the expansion rate decreases. The tube diameter continues to increase at the slower rate until it reaches its peak expansion of 38.7%. This change in slope of the expansion response is likely a result of the pressure waveform augmentation discussed in the previous section causing additional tube expansion in late systole. After reaching its peak, the tube begins to relax. Shortly after the beginning of relaxation there is a sharp momentary increase in tube diameter that signals the VAD outlet valve closure. After valve closure, the tube continues to relax to its end diastolic diameter.

### Downstream flow response

Fig 8 displays the downstream instantaneous centerline velocities over one normalized pump cycle for the rigid ($\omega_{Al\ pump}$ = 1235 RPM) and compliant ($\omega_{Al\ pump}$ = 1235 RPM, $P_{ch}$ = 103 mmHg) cases. Both response cycles are indexed to begin at the image corresponding to the first diastolic minimum after the Camera 2 trigger signal; the zero time instant defined in the Fig 8 corresponds to the zero time of the pressure signal shown in Fig 6. There are significant differences between the two response shapes. First, the centerline velocity, $v_{CL}$, plot of the rigid case has more pronounced local amplitude variations than does the plot of the compliant case.

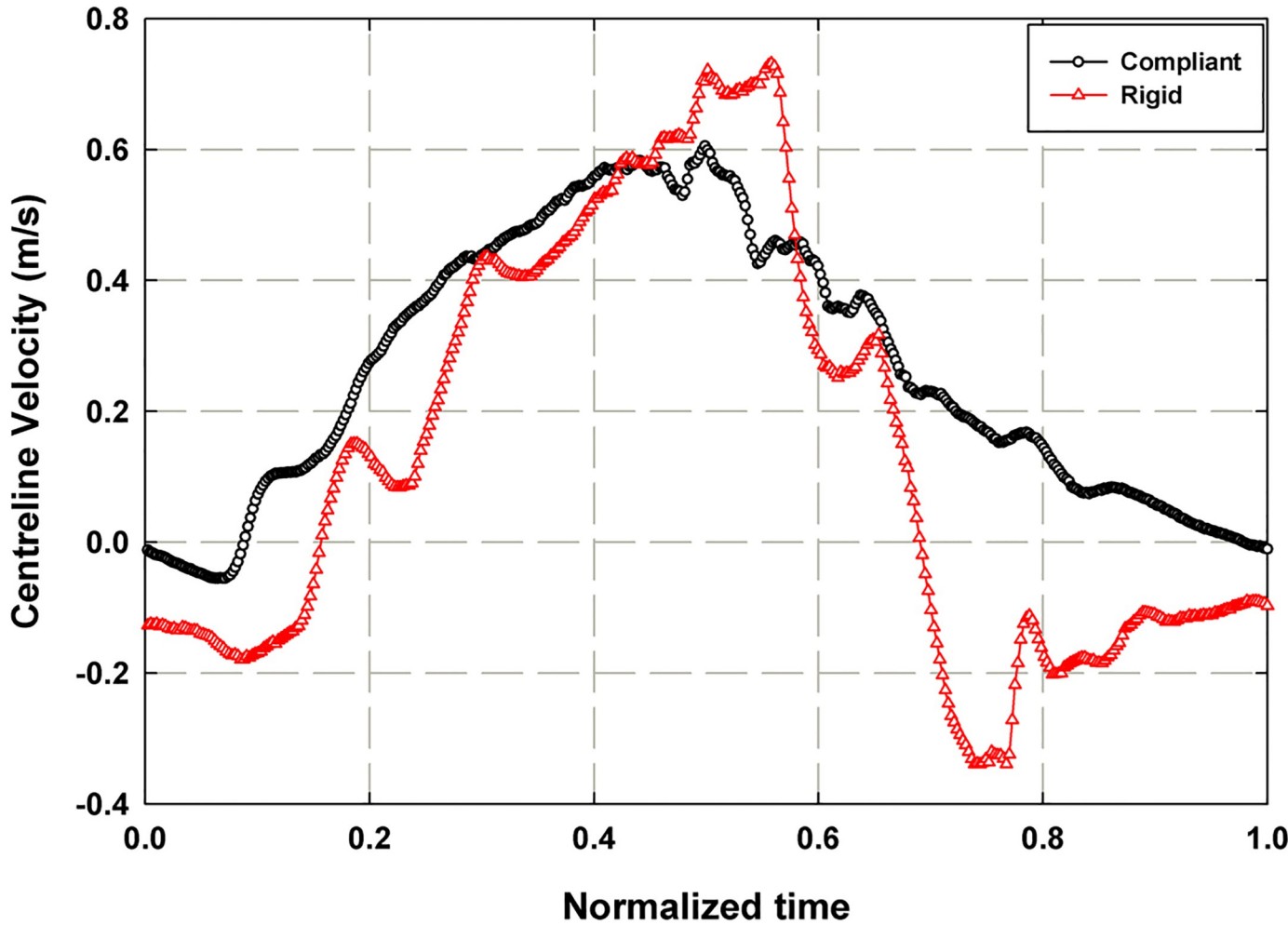

**Fig 8. Plot of centerline velocities ($v_{CL}$) for rigid ($\omega_{Al\ pump}$ = 1235 RPM) and compliant mock aorta ($\omega_{Al\ pump}$ = 1235 RPM, $P_{ch}$ = 103 mmHg) cases over one normalized pump cycle.**

This is likely a byproduct of the presence of multiple reflected pressure waves passing through the system. The rigid case also reaches both a higher positive and negative peak velocity during the cycle and reaches its positive peak velocity later in the cycle than does the compliant case. The increased reverse flow (negative velocity) in the rigid case indicates a high unsteady component in the pressure gradient driving the flow which is supported by the pressure waveform results. In the compliant case however, there is minimal reverse flow at the beginning of the cycle, indicating that the compliant tubing is having the effect of reducing unsteady effects in the downstream flow fields.

## Comparison of system performance parameters

To understand the interaction between the three parameters discussed in previous sections, it is useful to view the parameters together over one pump cycle. Fig 9 depicts the relationship between the normalized pressure, $P/P_{max}$, waveform and normalized downstream centerline velocity, $v_{CL}/v_{CL,max}$, over one normalized pump cycle obtained from the rigid case ($\omega_{Al\ pump}$ = 1235 RPM), where $P_{max}$ and $v_{CL,max}$ are the maximum pressure and centerline velocity over a cycle. The pressure waveform shows the occurrence of multiple reflected waves, the results of which are visible on the centerline velocity plot. The centerline velocity reaches its positive peak at the time of arrival of the third pressure peak, $t/\tau$ = ~0.55. It reaches its negative peak, that is nearly half the magnitude of the positive peak, at $t/\tau$ = ~0.75 and the velocity is negative for approximately half of the cycle time. The significant amount of reverse flow supports the observation that the downstream flow in the rigid case is substantially impacted by the unsteady effects of the pressure pulse from the VAD.

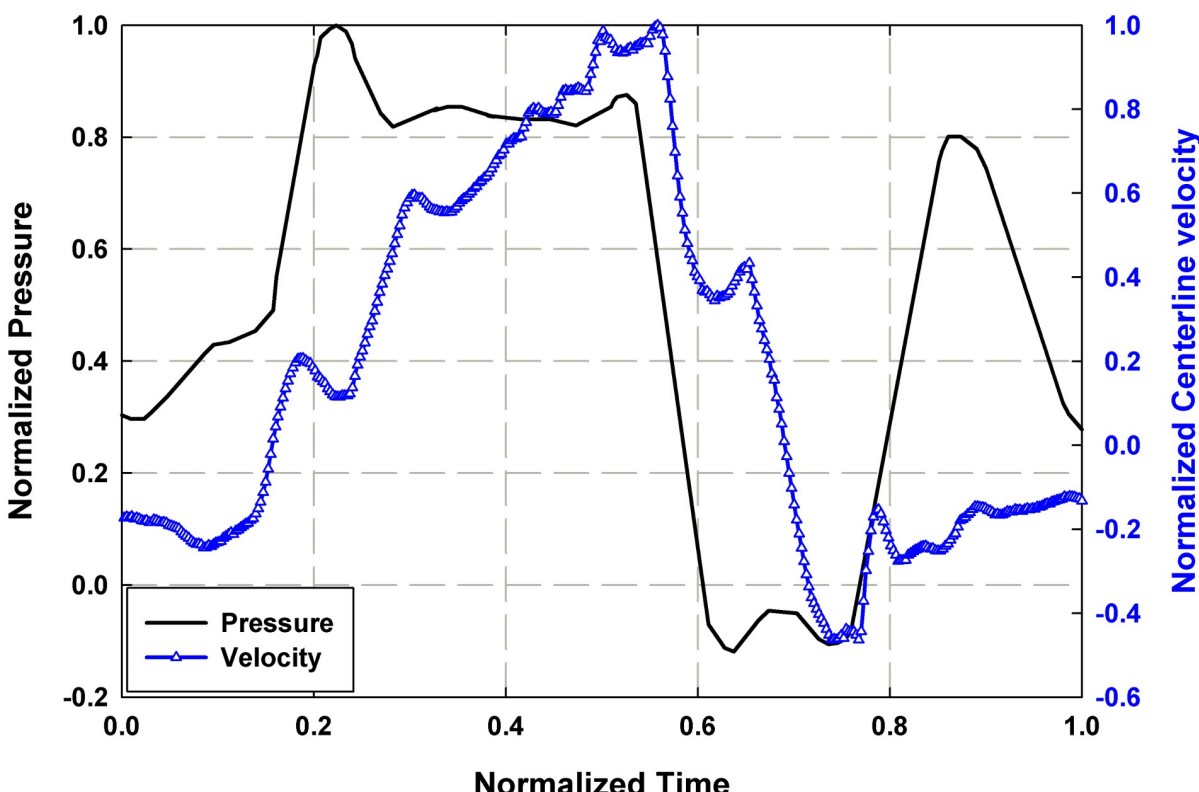

**Fig 9. Normalized plot of central aortic waveform ($P/P_{max}$) and downstream centerline velocity ($v_{CL}/v_{CL,max}$) for the rigid case with $\omega_{Al\ pump}$ = 1235 RPM.**

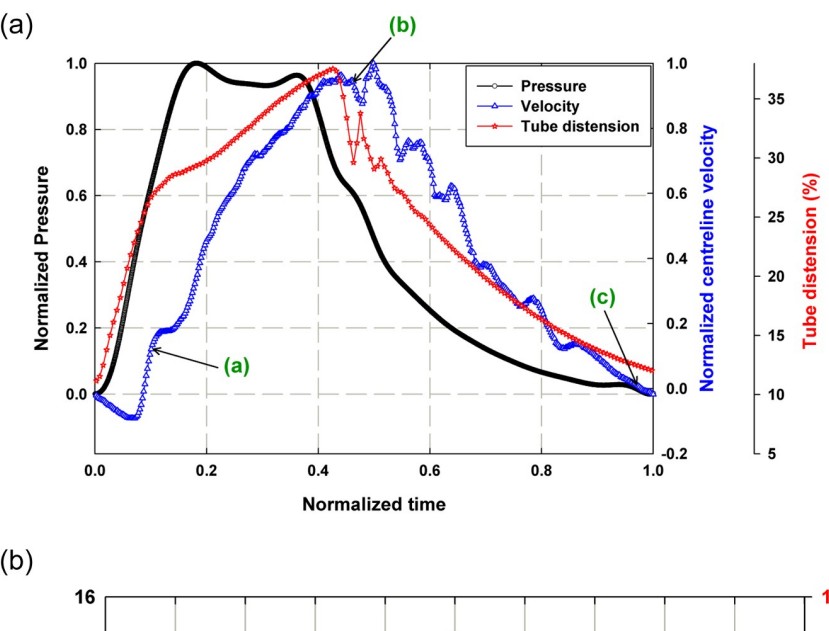

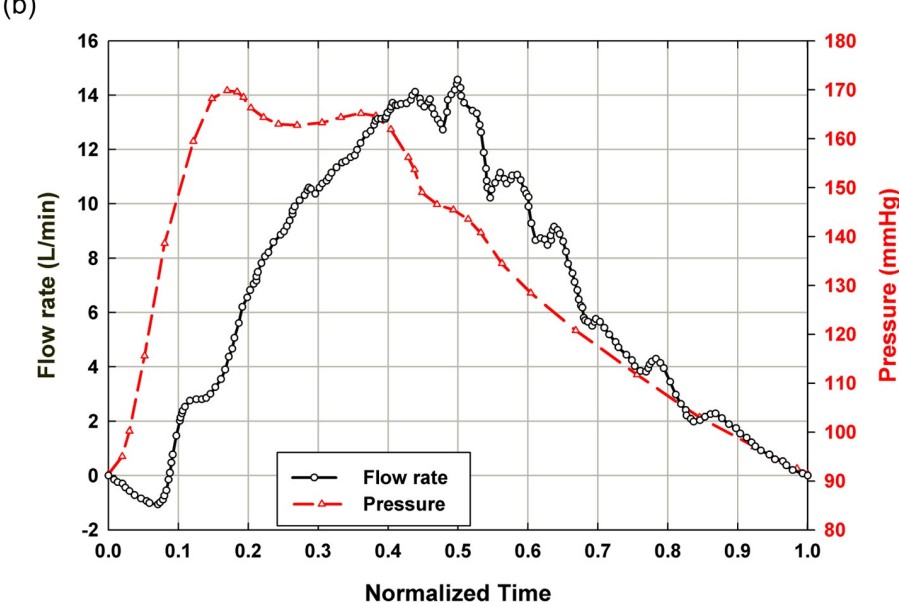

**Fig 10.** (a) Plot of normalized central aortic pressure waveform ($P/P_{max}$), normalized downstream centerline velocity ($v_{CL}/v_{CL,max}$) and normalized tube distension ($\Delta D/D \times 100$) and (b) plot of flow rate and the pressure for the compliant case with $P_{ch}$ = 103 mmHg and $\omega_{Al\ pump}$ = 1235 RPM. Points (a), (b) and (c) on Fig 10(a) are the time stamp of the 3 phases for which the velocity vector fields are plotted in Fig 5(a).

Fig 10(a) depicts the relationship between normalized pressure waveform, normalized downstream centerline velocity and normalized tube distension percentage, $\Delta D/D \times 100$, for one normalized pump cycle obtained from the compliant mock aorta case ($\omega_{Al\ pump}$ = 1235 RPM, $P_{ch}$ = 103 mmHg) where $D$ is the tube diameter. The initial tube expansion occurs in phase with the systolic pressure pulse and then slows down, reaching its maximum around the moment of arrival of the reflected pressure wave, at $t/\tau$ = ~0.41. The dicrotic notch can be identified on both the pressure and tube distension waveforms at $t/\tau$ = ~0.45. On the pressure waveform, it is identified as the inflection point after the peak of the reflected wave, and on the tube distension plot it is identified as the sharp drop and subsequent rise. The downstream centerline velocity response lags both the pressure and tube response, reaching its peak at $t/\tau$ =

~0.50. The centerline velocity is negative for approximately the first 10% of the cycle time, reaching a negative peak velocity that is 9% of the positive peak velocity at $t/\tau = 0.06$. The brief period of reverse flow at the beginning of the cycle is due to the time lag between pressure and velocity, likely attributable to the pressure decrease from the end of the previous cycle. This is a substantial improvement over the significant amount of reverse flow in the rigid case operated at the same afterload pump speed, which indicates that the compliant tube is serving to damp a significant amount of the unsteady behavior of the pump. The points (a), (b) and (c) on Fig 10(a) are the time stamp of the 3 phases during the initializing of the systole, during the systole around the maximum centerline velocity and at the diastole for which the velocity vector fields are plotted in Fig 5(a). Fig 10(b) shows the flow rate values, calculated from the velocity data, along with the pressure data for the compliant case. By integrating the flow rate curve over systolic ejection time, a stroke volume of 98.5 mL was obtained. Using the pulse frequency of 67.2 bpm (1.12 Hz), cardiac output was calculated to be 6.62 L/min.

## Impact of compliance on downstream flow steadiness

To investigate the pressure and flow responses of the system as a function of the resistance, or afterload, of the tubing network it is useful to explore the manner in which the transmission of flow pulsatility varies as a result of varying afterload conditions. In the rigid case, afterload conditions are determined solely by the speed of pump 2 (P2) and in the compliant case, the afterload conditions are determined by both the speed of P2 and by the distension of the mock aorta. Therefore, the transmission of pulsatility downstream varies depending on the speed of the afterload pump and on the compliance of the mock aorta.

Fig 11 depicts the changes in maximum positive and negative peak centerline velocity that occur as a result of increasing $\omega_{Al\ pump}$ for the rigid case. On the same figure, the absolute values of the ratio between the peak reverse flow during the cycle to the peak positive flow $|v_{pk,(-)}/v_{pk,(+)}|$ is presented. The absolute ratio reflects the increasing influence of the negative centerline velocity with the rigid tube as compared to the decreasing influence of the negative centerline velocity with the compliant tube. The positive peak centerline velocity is shown to decrease with increasing $\omega_{Al\ pump}$, while the negative peak centerline velocity remains relatively constant. As a result, the absolute value of the peak velocity ratio decreases with increasing $\omega_{Al\ pump}$. Therefore, for the rigid case, increasing the afterload pump speed increases the influence of reverse flow in the system.

Fig 12 plots changes in maximum positive and negative peak centerline velocity that occur with increasing values of mock aortic distensibility for the case where $P_{ch} = 103$ mmHg. Afterload pump (P2) speed also impacts the distensibility and hence each of the distensibility values correspond to a pump speed as reported in Table 1. As with the rigid case, the absolute ratio of peak negative to peak positive flow velocities is also displayed. At higher distensibility values, the peak negative flow and peak flow ratio approach zero and at the highest tested distensibility, they both reach zero. This demonstrates that the introduction of compliance into the system dampens the transmission of unsteady effects and produces near constant forward flow downstream, as the Windkessel theory suggests.

## Impact of compliance on pump energy

Table 1 presents a detailed breakdown of the experimental conditions, resulting distensibility values and per-minute energy output by the pump for the complete range of conditions undertaken in this study. The afterload pump speed varied from 980 RPM to 1540 RPM while two chamber pressure values applied in the compliant tube case. At the maximum tested afterload pump speed of 1540 RPM and the chamber pressure of 103 mmHg, the tube distended so

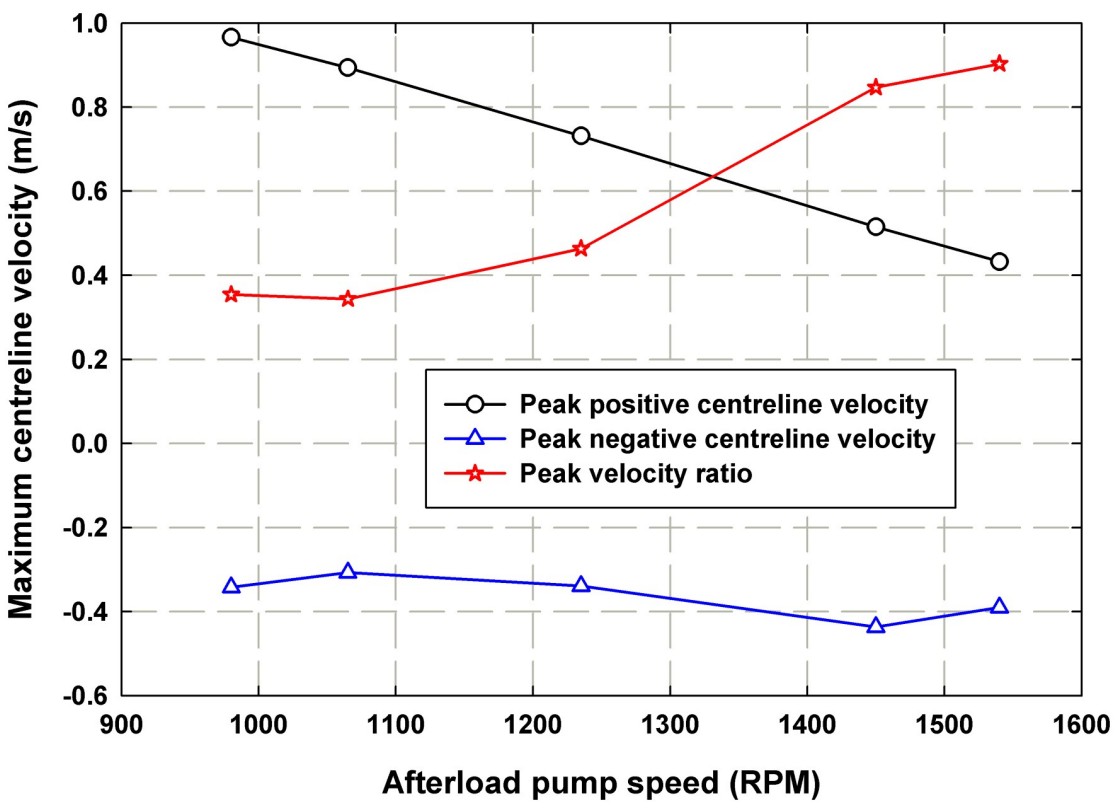

**Fig 11. Plot representing the peak positive downstream centerline velocity, peak negative centerline velocity and peak velocities ratio (peak negative to peak positive) as function of the afterload pump speed for the rigid case.**

much that it touched the inside walls of the pressure chamber, so the distensibility could not be properly measured. The energy values are obtained from Eq 4 and presented per minute rather than per ejection because, due to experimental conditions, both systolic ejection time and total cycle time vary based on $\omega_{Al\ pump}$ and $P_{ch}$. Therefore, looking at the per ejection energy alone would provide only a partial picture of how the pump energy varies based on experimental conditions.

Fig 13 summarizes the data in Table 1 of the impact of mock aorta distensibility ($d$) on the pump energy output for each of the tested $\omega_{Al\ pump}$ settings. It is shown that each $\omega_{Al\ pump}$ setting displays a similar trend. The rigid case ($d = 0$) places the highest energy demands on the pump for all cases by up to 33%. Once some degree of compliance is added, there is a substantial decrease in energy output by the pump. At a certain point after the initial decline in energy, there is a trend of a slight increase in energy output with the next increase in distensibility. This is most likely the point at which the flow supplied by the afterload pump ceases to be sufficient to fill the mock aorta, so the VAD requires more energy output to fill that extra volume. In addition, the inflection point where energy begins to rise shifts to lower distensibility values with decreasing $\omega_{Al\ pump}$ setting, indicating that with less afterload supply, more energy is required by the pump at lower distensibility values.

## Conclusion

This paper presented the results obtained using both a rigid tube and a compliant mock aorta in a mechanical flow loop analogous to the left side of a proposed working-mode capable

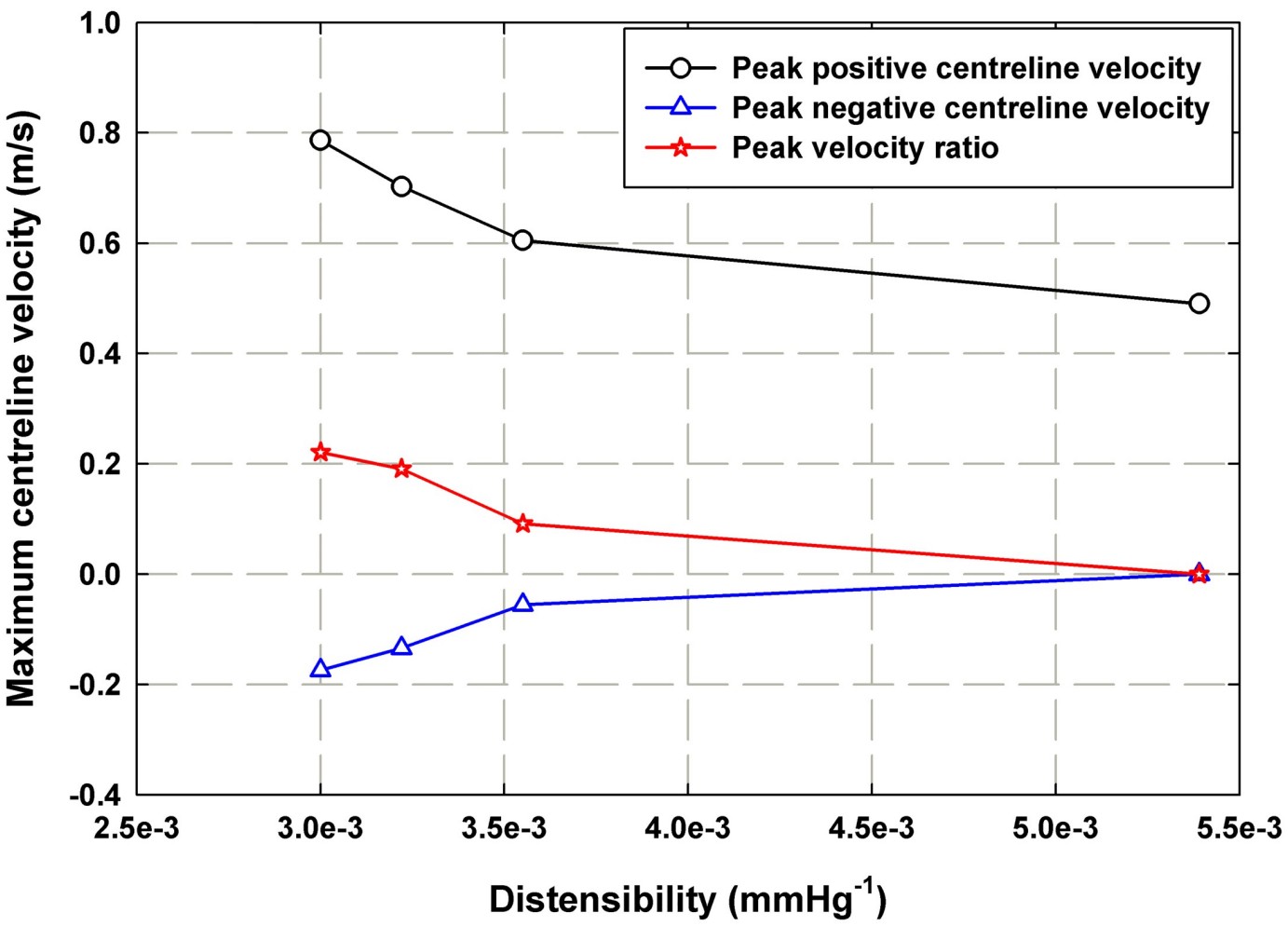

**Fig 12. Plot representing the peak positive downstream centerline velocity, peak negative centerline velocity and peak velocities ratio (peak negative to peak positive) as function of mock aorta distensibility for the compliant case where $P_{ch}$ = 103 mmHg.**

**Table 1. Summary of determined pump energy results.**

| $\omega_{Al\ pump}$ (RPM) | Chamber pressure, $P_{ch}$ (mmHg) | Distensibility, $d$ (mmHg$^{-1}$ × 10$^{-3}$) | Per-minute pump work (J/min) |
|---|---|---|---|
| 980 | Not applicable (Rigid case) | 0 | 129 |
| | 155 | 2.42 | 89 |
| | 103 | 3.00 | 87 |
| 1065 | Not applicable (Rigid case) | 0 | 126 |
| | 155 | 2.53 | 83 |
| | 103 | 3.23 | 84 |
| 1235 | Not applicable (Rigid case) | 0 | 111 |
| | 155 | 2.84 | 77 |
| | 103 | 3.55 | 82 |
| 1450 | Not applicable (Rigid case) | 0 | 84 |
| | 155 | 3.77 | 64 |
| | 103 | 5.39 | 69 |
| 1540 | Not applicable (Rigid case) | 0 | 70 |
| | 155 | 4.54 | 59 |
| | 103 | Not measured | Not measured |

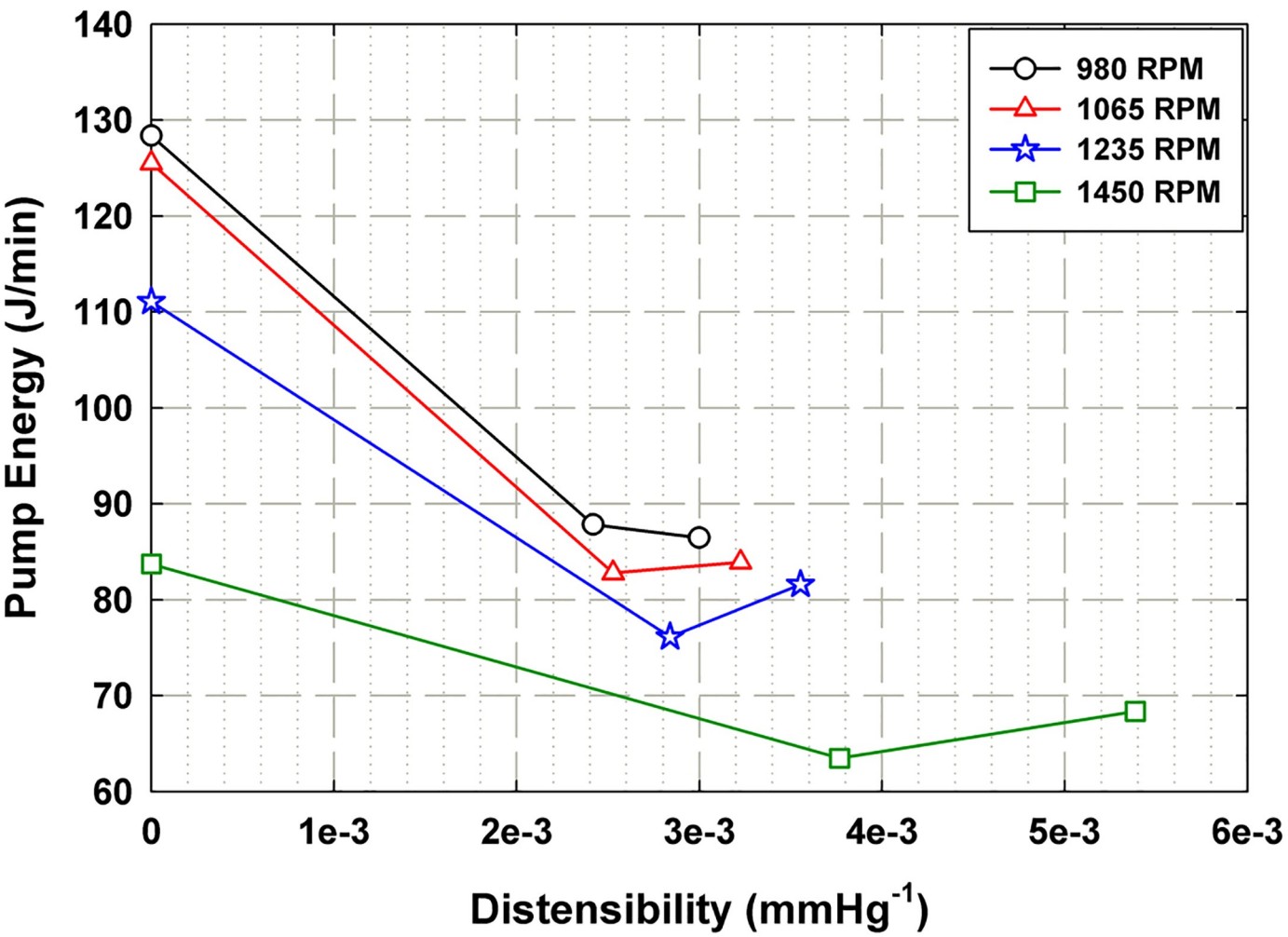

**Fig 13. Plot representing the impact of mock aorta distensibility ($d$) on the energy expenditure of the VAD per minute ($E_{VAD}$) for each tested setting of $\omega_{Al}$ $_{pump}$.**

EVHP system, using a VAD to simulate the necessary cardiovascular flow. This was done with the intent of assessing the need for a compliant mock aorta in the proposed EVHP system based on the knowledge that aortic compliance has a significant impact on cardiac performance *in-vivo*. The impact of compliance was presented based on the central aortic waveform shape, the tube distension response and the downstream flow response. This culminated in an analysis of the impact of mock aortic distensibility on the transmission of unsteady effects downstream and on pump energy requirements for the full range of tested experimental conditions.

Results indicate that the addition of a compliant mock aorta promotes better system performance based on all measured parameters. In all tested cases, the introduction of compliance smoothed the central aortic pressure waveforms making them closely resemble expected physiological shapes. As well, fluctuations in the unsteady flow downstream were dampened. The downstream velocity results demonstrate a lower peak velocity ratio than the rigid tube cases, supporting the conclusion that the compliant mock aorta serves to dampen the pressure pulse transmission through the system and create quasi-steady flow downstream. Most importantly, this study proves that rigid tubing in place of an aorta *ex-vivo* places a much larger workload

on the pump than does a compliant tube under the same set of experimental conditions. For the range of tested experimental conditions, the pump energy output per minute was up to 33% higher in rigid cases compared to the compliant cases. This provides experimental verification of the Windkessel theory and supports the need for a compliant mock aorta in the proposed EVHP technology capable of near-physiologic working mode.

## Nomenclature

The unit of pressure used in this paper is millimeters of mercury (mmHg), the standard unit used for blood pressure measurement. The conversion factor from mmHg to Pa is 133.32 Pa/mmHg.

## Author Contributions

**Conceptualization:** Darren H. Freed, David S. Nobes.

**Data curation:** Katie Cameron, Mouhammad El Hassan, Reza Sabbagh.

**Formal analysis:** Katie Cameron, Reza Sabbagh.

**Funding acquisition:** Darren H. Freed, David S. Nobes.

**Investigation:** Katie Cameron, Reza Sabbagh.

**Methodology:** Katie Cameron, David S. Nobes.

**Project administration:** David S. Nobes.

**Software:** David S. Nobes.

**Supervision:** Darren H. Freed, David S. Nobes.

**Validation:** Katie Cameron, Mouhammad El Hassan, Reza Sabbagh.

**Visualization:** Katie Cameron.

**Writing – original draft:** Katie Cameron.

**Writing – review & editing:** Mouhammad El Hassan, Reza Sabbagh, Darren H. Freed, David S. Nobes.

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
