## [Decision Letter · Decision Letter 0]

23 Apr 2020

PONE-D-20-03379

Experimental investigation into the effect of compliance of a mock aorta on cardiac performance

PLOS ONE

Dear Prof. Nobes,

Thank you for submitting your manuscript to PLOS ONE. After careful consideration, we feel that it has merit but does not fully meet PLOS ONE’s publication criteria as it currently stands. Therefore, we invite you to submit a revised version of the manuscript that addresses the points raised during the review process.

Overall both reviewers are positive but feel that the presentation of material and provided analysis can be improved (see both reviewers' comments. Some of the details such as material properties, Womersely number, etc which are necessary for completeness are missing (Reviewer 1). Reviewer 2 also suggest additional analysis of the data and providing the flow curve. Finally, based on PLOS ONE criteria, the underlying data should be publicly available (through supplementary materials or a public repository). 

We would appreciate receiving your revised manuscript by Jun 07 2020 11:59PM. To enhance the reproducibility of your results, we recommend that if applicable you deposit your laboratory protocols in protocols.io, where a protocol can be assigned its own identifier (DOI) such that it can be cited independently in the future. For instructions see: http://journals.plos.org/plosone/s/submission-guidelines#loc-laboratory-protocols

We look forward to receiving your revised manuscript.

Kind regards,

Iman Borazjani, Ph.D.

Academic Editor

PLOS ONE

"Financial support for this project was provided by the Natural Sciences and

Engineering Research Council (NSERC) of Canada, the Canadian Foundation of

Innovation (CFI), the Canadian National Transplant Research Program (CIHR/CNTRP)

and the University Hospital Foundation (UHF)."

"Financial support for this project was provided by the Natural Sciences and Engineering Research Council (NSERC) of Canada."

Reviewers' comments:

Reviewer's Responses to Questions

**Comments to the Author**

1. Is the manuscript technically sound, and do the data support the conclusions?

Reviewer #1: Yes

Reviewer #2: Partly

2. Has the statistical analysis been performed appropriately and rigorously? 

Reviewer #1: Yes

Reviewer #2: No

3. Have the authors made all data underlying the findings in their manuscript fully available?

Reviewer #1: Yes

Reviewer #2: No

4. Is the manuscript presented in an intelligible fashion and written in standard English?

Reviewer #1: Yes

Reviewer #2: Yes

5. Review Comments to the Author

Reviewer #1: This paper investigates the effects of a rigid tube and a compliant mock aorta placed in the left side of an ex-vivo heart perfusion (EVHP) on pressure waveforms, centerline velocity, tube distension, and energy demand on the pump. It concludes that given the healthier physiologic pressure waveforms, steadier downstream flow, and reduced energetic demands on the pump for the compliant mock aorta, it provides the experimental verifications of Windkessel theory and suggests the use of mock aorta in the EVHP system.

Comments:

I believe this is an interesting study since it provides experimental evidence on the effects of elastic aorta model compared to a rigid one. The value of this paper from fluid mechanics perspective justifies its publication after some minor issues are addressed:

1. The discussion on the cardiac performance can be extended to other parameters rather than pump energy output, in order to better evaluate the healthier physiological condition. Parameters such as ejection fraction or h-efficiency (Song & Borazjani, 2015) can be helpful.

2. It does not seem necessary to talk about the Womersley number to provide the reasoning for using the unsteady Bernoulli equations; however, the authors did not provide the Womersley number related to their own experiments to justify the use of unsteady Bernoulli equation. In addition, the first sentence of second paragraph of page 6 is too general and is not a fact. Also, the reference number 28 suggested a limit value of α=1 not 10.

3. It seems that friction losses should be included in Eq. 2 as it is included in Eq. 3. Since the authors choose to start from general form (Eq. 2), it would be better to provide all the simplification and the underlying reasons for them. For example, it might not be clear for the readers how the fourth term in Eq. 2 is neglected.

4. The structural properties of the material employed to make the mock aorta is not addressed. Please provide the material properties in the manuscript or refer to appropriate literature.

5. It might be helpful to better present the experimental setup by clarifying the differences between the current experimental setup (Fig 4) and EVHP system (Fig 2) at the beginning of the experimental setup section.

6. For the averaging of the centerline velocity, to be clearer, it is better to first present the underlying reason and then providing the averaging methodology.

7. It is recommended to add a new section for pump energy calculations rather than having it under the experimental setup section.

8. In Fig. 6, it is not clear how the negative values of diastolic pressure for rigid tube is measured; however, in the manuscript, the diastolic pressure is reported 68.1 for this case, which does not consider the negative parts of the pressure waveform and is inconsistent with the definition of diastolic pressure presented on page 6.

9. In Fig. 7, horizontal axes must be shifted 0.5 of the normalized time to fix its zero value to the beginning of the systole. In addition, the sharp momentary increase after relaxation is not clear in the plot. Also, please add the appropriate legends to Fig.7 to distinguish the contributors to the tube expansion.

10. The description of the indexing cycles for Fig. 8 is not clear. Does the zero time instant set to be consistent with the pressure plots, as they are plotted together at later figures. Please clarify this sentence: “Both response cycles are indexed to begin at the image corresponding to the first diastolic minimum after the Camera 2 trigger signal.”

11. In fig. 9, Normalized pressure does not show negative value; however its absolute values does (Fig.6). In addition, pressure values at the beginning and end of the cycle do not match. Please clarify/fix it.

12. The dicrotic notch for pressure plot of Fig 10 is reported as 0.45, which is not consistent with the previous value of 0.4 on page 16 and the definition of dicrotic notch presented on page 6.

13. It is better to use the absolute value of the peak velocity ratio in Fig. 11, in order to visually show the increasing influence of negative centerline velocity, i.e., reverse flow.

14. Why there is a jump in data provided for Fig 12? Is there any experimental difficulty in gathering the data for this interval?

Some other comments:

1. Please revise the last sentence of the second paragraph on page 4. It seems that both impacts are basically the same and it strike me as a repetition: the impact of the lack of the elastic response and the impact of adding a compliant aorta.

2. There are minor typos and inconsistencies in references such as reference number 27.

3. The cross-reference to table 1 on page 21 needs to be corrected.

Reviewer #2: The study addresses an important problem. While the authors have done a good job with the study and composed the manuscript well, the current state of the manuscript needs significant improvement. The manuscript could use some level of reorganization, rewriting so that it is clearer to the reader. Several data acquisition campaigns have been undertaken, but the resulting data analysis could be improved, the analysis of the high-fidelity data acquired (for example the PIV) could be analyzed in a more in-depth manner.

The following are some specific comments:

Please present the flow rate curve along with the pressure curve. Getting the right cardiac output is important.

How about PIV data during systole? What is the centerline velocity during systole?

PIV uncertainty estimation has been conducted, but only limited number of factors are considered. See Raffel et al (2013) or Raghav et al (2018) for more details.

Fig 5 please mention exact time point during diastole? The positive velocity near the wall region is unrealistic, if r/D =0 and 1 are the walls

With regards to the unsteady Bernoulli formulation, please discuss how it compares to Donati et al Circulation (2017).

Why were these specific values of distensibility chosen? Fig 13 and 12 - the x axis is labeled differently, please use same notations.

6. PLOS authors have the option to publish the peer review history of their article (what does this mean?). If published, this will include your full peer review and any attached files.

Reviewer #1: No

Reviewer #2: No

---

## [Author Response · Author response to Decision Letter 0]

19 Jun 2020

Our comments addressing comments from the reviewers have been attached as a WORD docx.

---

## [Decision Letter · Decision Letter 1]

16 Jul 2020

PONE-D-20-03379R1

Experimental investigation into the effect of compliance of a mock aorta on cardiac performance

PLOS ONE

Dear Dr. Nobes,

Thank you for submitting your manuscript to PLOS ONE. After careful consideration, we feel that it has merit but does not fully meet PLOS ONE’s publication criteria as it currently stands. Therefore, we invite you to submit a revised version of the manuscript that addresses the points raised during the review process.

As you can see Reviewer#1 is satisfied with your response and has only a few minor comments. However, reviewer#2 brings up major concerns about the soundness of presented results, .e.g,, measured velocities show higher values near walls (Fig. 5) or the shown pressure and flow curve of the system does not agree with the physiological curve (Fig. 10), among others.  The revision has to be made above and beyond the reviewers comments to ensure clarity and correctness of presented results. In addition, the presented results/conclusions are for the given mock loop and should not be generalized. In fact, in another mock loop with different pump, resistance, or capacitance elements a rigid aorta may produce a physiologic waveform. In general,  the metrics used for comparison (pressure, flow rate, etc) are more sensitive to other factors (e.g., pump, loop, etc) than the one investigated (e.g., compliance) a sensitivity study is required or the limitations needs to be clearly discussed.  

We look forward to receiving your revised manuscript.

Kind regards,

Iman Borazjani, Ph.D.

Academic Editor

PLOS ONE

Reviewers' comments:

Reviewer's Responses to Questions

**Comments to the Author**

1. If the authors have adequately addressed your comments raised in a previous round of review and you feel that this manuscript is now acceptable for publication, you may indicate that here to bypass the “Comments to the Author” section, enter your conflict of interest statement in the “Confidential to Editor” section, and submit your "Accept" recommendation.

Reviewer #1: All comments have been addressed

Reviewer #2: (No Response)

2. Is the manuscript technically sound, and do the data support the conclusions?

Reviewer #1: Yes

Reviewer #2: No

3. Has the statistical analysis been performed appropriately and rigorously? 

Reviewer #1: Yes

Reviewer #2: N/A

4. Have the authors made all data underlying the findings in their manuscript fully available?

Reviewer #1: Yes

Reviewer #2: No

5. Is the manuscript presented in an intelligible fashion and written in standard English?

Reviewer #1: Yes

Reviewer #2: Yes

6. Review Comments to the Author

Reviewer #1: The authors have responded to the comments and suggestions with sufficient detail and the paper is in publication-mode. However there are two minor points that need to be addressed:

1. In the extended discussion regarding the cardiac performance, the added reference discusses a new metric (hydrodynamic efficiency) for cardiac performance and shows that it illustrates more details than ejection fraction; therefore, it is not a good reference for the importance of ejection fraction.

2. In equations 2 and 3, density is missing in the acceleration terms.

Reviewer #2: - New figure 5b that has been provided does not make sense. How come the velocity near the walls are much higher than the velocity near the center? During systolic phase (flow acceleration) the bulk flow should accelerate and then decelerate, with the maximum velocity nominally close to centerline. Which phase (time point) of systole is presented?

- Fig 10b - the pressure profile and flow rate profile are not physiological, when compared to expected aortic pressure. Please mark when systolic phase ends? How does flow rate not go down to zero at the end of systole? It slowly decays to zero. After systole, the flow rate has to be close to zero for the entirety of diastole, which is not being simulated here.

- And in addition, the end of normalized time (1.0) does not continue with start of normalized time (0), this is a major discrepancy in the presented data.

- Fig 10a and 10b - do not provide the same information for both rigid and compliant case (normalized pressure or absolute pressure), providing a mix, does not allow for a fair comparison.

- How does centerline velocity show fluctuations when the corresponding flow rate profile does not have that many fluctuations?

7. PLOS authors have the option to publish the peer review history of their article (what does this mean?). If published, this will include your full peer review and any attached files.

Reviewer #1: No

Reviewer #2: No

---

## [Author Response · Author response to Decision Letter 1]

14 Aug 2020

A details file responding to the reviewer comments has been uploaded in the Attached Files section of this reply.

---

## [Decision Letter · Decision Letter 2]

10 Sep 2020

Experimental investigation into the effect of compliance of a mock aorta on cardiac performance

PONE-D-20-03379R2

Dear Dr. Nobes,

We’re pleased to inform you that your manuscript has been judged scientifically suitable for publication and will be formally accepted for publication once it meets all outstanding technical requirements.

Kind regards,

Iman Borazjani, Ph.D.

Academic Editor

PLOS ONE

Additional Editor Comments (optional):

Reviewers' comments:

Reviewer's Responses to Questions

**Comments to the Author**

1. If the authors have adequately addressed your comments raised in a previous round of review and you feel that this manuscript is now acceptable for publication, you may indicate that here to bypass the “Comments to the Author” section, enter your conflict of interest statement in the “Confidential to Editor” section, and submit your "Accept" recommendation.

Reviewer #2: All comments have been addressed

2. Is the manuscript technically sound, and do the data support the conclusions?

Reviewer #2: Partly

3. Has the statistical analysis been performed appropriately and rigorously? 

Reviewer #2: N/A

4. Have the authors made all data underlying the findings in their manuscript fully available?

Reviewer #2: No

5. Is the manuscript presented in an intelligible fashion and written in standard English?

Reviewer #2: Yes

6. Review Comments to the Author

Reviewer #2: (No Response)

7. PLOS authors have the option to publish the peer review history of their article (what does this mean?). If published, this will include your full peer review and any attached files.

Reviewer #2: No

---

## [Editor Report · Acceptance letter]

30 Sep 2020

PONE-D-20-03379R2 

Experimental investigation into the effect of compliance of a mock aorta on cardiac performance 

Dear Dr. Nobes:

I'm pleased to inform you that your manuscript has been deemed suitable for publication in PLOS ONE. Congratulations! Your manuscript is now with our production department. 

Kind regards, 

on behalf of

Dr. Iman Borazjani 

Academic Editor

PLOS ONE